# Sea Cucumber *(Isostichopus badionotus*): Bioactivity and Wound Healing Capacity In Vitro of Small Peptide Isolates from Digests of Whole-Body Wall or Purified Collagen

**DOI:** 10.3390/md23110411

**Published:** 2025-10-22

**Authors:** Leticia Olivera-Castillo, George Grant, Oscar Medina-Contreras, Honorio Cruz-López, Leydi Carrillo-Cocom, Ariadnna Cruz-Córdova, Frank Segura-Cadiz, Daniel Alejandro Fernández-Velasco, Sergio Rodríguez-Morales, Juan Valerio Cauich-Rodríguez, Rosa Esther Moo-Puc, César Puerto-Castillo, Gabriela de Jesus Moo-Pech, Jonatan Jafet Uuh-Narvaez, Miguel Angel Olvera-Novoa, Rossanna Rodriguez-Canul

**Affiliations:** 1Departamento Recursos del Mar, Centro de Investigación y de Estudios Avanzados del IPN, Unidad Mérida (Cinvestav), Mérida 97310, Mexico; fscadiz@gmail.com (F.S.-C.); cesarp@cinvestav.mx (C.P.-C.); gabriela.moojp@gmail.com (G.d.J.M.-P.); miguel.olvera@cinvestav.mx (M.A.O.-N.); rossana.rodriguez@cinvestav.mx (R.R.-C.); 2Independent Researcher, Aberdeen AB21 7AU, UK; 3Hospital Infantil de México Federico Gómez, Ciudad de México 06720, Mexicoariadnnacruz@yahoo.com.mx (A.C.-C.); 4Escuela Nacional de Estudios Superiores Unidad Mérida (ENES Mérida), Universidad Nacional Autónoma de México, Ciudad de México 04510, Mexico; honorio@ciencias.unam.mx; 5Facultad de Ingeniería Química, Universidad Autónoma de Yucatán, Mérida 97203, Mexico; leydi.carrillo@correo.uady.mx (L.C.-C.); jonatan.uuh@correo.uady.mx (J.J.U.-N.); 6Laboratorio de Fisicoquímica e Ingeniería de Proteínas, Departamento de Bioquímica, Facultad de Medicina, Universidad Nacional Autónoma de México, Ciudad de México 04510, Mexico; fdaniel@unam.mx; 7Unidad de Química en Sisal, Facultad de Química, Universidad Nacional Autónoma de México, Ciudad de México 04510, Mexico; sergiorodriguez@unam.mx; 8Centro de Investigación Científica de Yucatán, Unidad de Materiales, Mérida 97205, Mexico; jvcr@cicy.mx; 9Secretaría de Ciencia, Humanidades, Tecnología e Innovación—Hospital Regional de Alta Especialidad de la Península de Yucatán, IMSS-Bienestar, Mérida 97130, Mexico; rosa.moo@secihti.mx; 10Ciencia y Herbolaria, San Jose Kuche, Conkal 97345, Mexico

**Keywords:** Holothuroid, body wall proteins, collagen, bioactive peptides, small molecular weight peptides, wound healing in vitro, antioxidant activity

## Abstract

Low-molecular-weight peptides derived from the digestion of body wall proteins in some sea cucumber species have wound-healing and health-promoting properties, but their potency varies widely by species, growth environment, age, and season. For the first time, small peptide (1–3 kDa) fractions have been isolated from the whole-body wall of the sea cucumber *Isostichopus badionotus* and its constituent collagen and tested for wound healing capacity in vitro. Ultrafiltered digests (1–3 kDa) of the pure collagen, as well as 1–3 kDa digests of the whole-body wall, had potent antioxidant activities and promoted rapid wound healing in a keratinocyte scratch wound assay. Gene expression studies suggested that the wound-healing actions of the individual collagen and body wall 1–3 kDa fractions differed significantly. Low-molecular-weight peptides derived from *I. badionotus* collagen did promote wound healing in vitro; however, their efficacy may have been modulated by additional factors produced during body wall or collagen digestion. These findings need to be confirmed in vivo.

## 1. Introduction

Sea cucumbers (Holothuroidea: Echinodermata) are benthic marine animals found in intertidal zones and shallow environments of the sea floor. Worldwide, at least 1770 species of sea cucumbers are known, of which around 80 are intensively fished and harvested for use as health-promoting foodstuffs or as sources of nutraceuticals [1,2,3].

Sea cucumbers can fully regenerate after damage to or loss of internal organs, including the digestive tract [4,5]. For this reason, they are a source of many metabolism-modulating and potentially health-promoting factors, such as saponins, glycosaminoglycans, chondroitin sulfate, sulfated polysaccharides, fucoidan, phenolics, lectins, cerebrosides, sterols, and bioactive peptides [6,7,8]. The content and potency of these constituents can vary significantly depending on species, growth environment, physiological/stress state, and handling or processing of captured or harvested animals [9,10,11].

Many recent studies have centred on bioactive peptides produced during the digestion of sea cucumber body wall proteins, including collagen, the main proteinaceous component of the body wall. Reports indicate that these peptides have anticancer, antihypertensive, immune-enhancing, brain-modulating, and wound-healing properties in vitro and in vivo [7,8,12,13].

Sea cucumber *Isostichopus badionotus* is distributed primarily in the Caribbean Sea and along the Western Atlantic Ocean coast, including off the Yucatan Peninsula of Mexico. Studies have shown that this species’ body wall contains bioactive factors, including glycosaminoglycans, fucoidins, and proteinaceous components that can ameliorate or block inflammation and modulate cellular metabolism in potentially beneficial health-promoting ways, both in vitro and in vivo [8,14,15].

The health-modulating properties of low-molecular-weight, or small, peptides from *I. badionotus* have not been evaluated to date. In the present study, collagen from *I. badionotus* was isolated and characterized. Then, 1–3 kDa fractions of digests of pure collagen and whole-body wall were collected and tested for bioactivity and the ability to promote wound healing in a scratch wound assay.

## 2. Results

### 2.1. Composition of I. badionotus

The moisture content of the skin-free body wall of *I. badionotus* was 800 g kg^−1^ wet body weight. On a dry weight basis, its ash content was 400 g kg^−1^, lipid content was 200 g kg^−1^, and protein content was 330 g kg^−1^. Collagen comprised 50 ± 10% of body wall protein, and 60% of that collagen was in the form of intact fibrils.

### 2.2. Initial Screening of Body Wall Fractions for Wound Healing Activity

Low-molecular-weight compounds produced during enzymatic digestion of sea cucumber body wall are reported to have anti-inflammatory and wound-healing properties in vitro and in vivo [7,8]. The level and potency of these factors vary significantly depending on the species, region, environment, growth conditions, and storage and handling after collection. To establish whether enzymatic digests of *I. badionotus* captured off the coast of Yucatan, Mexico, possessed significant wound-healing properties, an initial study was conducted in which three ultrafiltrate fractions were obtained from enzymatic (papain) digestion of the body wall (BW): <1 kDa BW, 1–3 kDa BW and >3 kDa BW. This initial wound healing evaluation indicated that the 1–3 kDa ultrafiltrate fraction of digested *I. badionotus* skin-free body wall (1–3 kDa BW) promoted wound healing over 24 h in a scratch wound assay based on HaCat cells (Figure 1a). In contrast, wound closure was incomplete on control medium plates as well as plates treated with the <1 kDa BW or >3 kDa BW fractions.

The 1–3 kDa BW ultrafiltrate was active in both ABTS and ORAC antioxidant assays (Figure 1c). However, the potencies differed significantly, with the Trolox equivalent activity in the ORAC assay being approximately four-fold higher than in the ABTS assay.

The antibacterial capacity of the *I. badionotus* preparation (1–3 kDa BW) was tested against 5 × 10^5^ CFU of *Escherichia coli* ATCC 25922, *Pseudomonas aeruginosa* ATCC 13637, *Pseudomonas aeruginosa* 64D (MDR), *Staphylococcus aureus* ATCC 25725, *Staphylococcus aureus* MR (Methicillin resistant), *Enterococcus faecalis* ATCC 51299 (Vancomycin resistant) and *Enterococcus faecium* 683D (vancomycin-resistant). Following incubation for 18 h at 37 °C, no bacterial growth was observed with any of the tested bacterial species cultured in 4 mg/mL 1–3 kDa BW, variable growth was evident with 2 mg/mL 1–3 kDa BW, while expected growth occurred at 1–3 kDa BW concentrations of ≤1 mg/mL.

Ten µL of the bacterial suspensions that had been incubated for 18 h in 4 mg/mL 1–3 kDa BW were then plated out on Blood agar or Luria Bertani agar and incubated at 37 °C for 18 h to evaluate viability. Significant growth was evident with *Staphylococcus aureus* ATCC 25725, *Staphylococcus aureus* MR (Methicillin-resistant), *Enterococcus faecalis* ATCC 51299 (Vancomycin-resistant) and *Enterococcus faecium* 683D (Vancomycin-resistant). In contrast, no growth was evident with *Escherichia coli* ATCC 25922, *Pseudomonas aeruginosa* ATCC 13637, or *Pseudomonas aeruginosa* 64D (MDR). These results indicate that, at a concentration of 4 mg/mL, *I. badionotus* 1–3 kDa BW peptides were bacteriostatic against Gram-positive bacteria, but bactericidal to the Gram-negative bacteria tested.

### 2.3. Collagen Isolation and Characterization

Collagen is the principal protein component of the sea cucumber body wall. As a result of our preliminary finding that skin-free *I. badionotus* body wall digest had wound healing capacity and 50 ± 10% of its body protein was collagen [two-thirds as intact fibrils], this body wall constituent was isolated by pepsin-digestion and characterized.

#### 2.3.1. Scanning Electron Microscope

Scanning electron microscope (SEM) images of the collagen samples obtained by pepsin-digestion of *I. badionotus* show well-developed fibril networks with thin, relatively uniform, and densely interwoven fibrils of collagen consistent with a homogeneously clustered network (Figure 2).

#### 2.3.2. SDS-PAGE

Electrophoresis (SDS-PAGE) revealed that collagen isolates from *I. badionotus* body wall consisted primarily of an α chain [~131 kDa] but also contained small amounts of a β chain [~220 kDa] (Figure 3). No small-molecular-weight impurity bands were evident, indicating that the molecular structure of the collagen was not destabilized or degraded during processing and extraction.

#### 2.3.3. Amino Acid Composition

The amino acid composition of the collagen extracted from *I. badionotus* body wall shows glycine (Gly) to be the most abundant amino acid, followed by glutamic acid (Glu), alanine (Ala), hydroxyproline (Hyp), proline (Pro), and arginine (Arg) (Table 1). Histidine (His), methionine (Met), hydroxylysine (Hyl), lysine (Lys), isoleucine (Ile), leucine (Leu), and tyrosine (Tyr) were present in low content, and cysteine (Cys) was present in trace amounts. The imino acids (Pro + Hyp) comprised 17.2 per cent of residues in the collagen preparation.

#### 2.3.4. UV–Visible Spectra and Fourier Transform Infrared Spectroscopy

Collagen extracted from *I. badionotus* body wall showed a strong primary absorption band at 226 nm (Figure 4a), similar to fish collagen (*Totoaba macdonaldi*). There was an additional small absorption band at 258 nm, corresponding to amino acids with aromatic rings (Phe and Tyr), indicating a low content of these amino acids. This finding aligns with the amino acid profile and further confirms the purity of the extracted collagen.

Collagen extracted from *I. badionotus* body wall and fish collagen showed similar and characteristic absorption bands in FTIR spectroscopy (Figure 4b). These were assigned as amide A (3295 cm^−1^), amide B (2933 cm^−1^), amide I (1636 cm^−1^), amide II (1543 cm^−1^), and amide III (1234 cm^−1^). These bands are comparable with the FTIR spectra of collagen from marine species; amide A represents the stretching vibrations of N–H and C–H, and amide B that of –NH. Amide I represents C = O stretching vibrations coupled with N–H bending vibrations, CN stretching, and CCN deformation. The presence of amide I bands in the 1600–1660 cm^−1^ range indicates the presence of a triple-helical structure. The bending vibration of N–H coupled to C–N is represented by the amide II band, which predicts the characteristic peaks of protein secondary structures. The amide III band indicates a complex mix of α-helices and β-sheets along with a random coil of the protein structure. Furthermore, the ratio between amide III and pyrrolidine ring (band at 1450 cm^−1^) vibration of proline and hydroxyproline was around 1.06, indicative of a triple helical structure for collagen from *I. badionotus* body wall.

#### 2.3.5. Circular Dichroism and X-Ray Diffraction

Circular dichroism (CD) is an efficient spectroscopy technique for confirming the structural integrity of the collagen triple helix. The collagen from *I. badionotus* body wall exhibited a weak positive absorption peak at 220 nm, and a negative one at 195 nm (Figure 5a). This indicates it is similar to that of other sea cucumber species and in line with the FTIR spectra (Figure 4b); that is, it had an intact triple helical structure. Temperature-induced unfolding of this structure exhibited a monophasic change in ellipticity at 222 nm (Figure 5b). The estimated denaturation temperature (Td) for collagen from *I. badionotus* body wall was 32.5 ± 0.2 °C. In sea cucumber collagens, Td is variable, but the value recorded for *I. badionotus* was at the high end of the range (32.3 to 34.6 °C). Elevated Tds have been linked to high imino acid content in collagens. In the present study, *I. badionotus* collagen had an imino acid content of 172 residues/1000 residues, which is comparable to that for collagens of similar thermal stability. By contrast, sea cucumber collagens with lower imino acid contents have Tds as low as 17.9 °C (Appendix A).

The *I. badionotus* collagen XRD pattern exhibits clear reflections corresponding to an amorphous natural polymer (protein) as indicated by the presence of two peaks similar to the fish (*T. macdonaldi*) collagen (Figure 5c). One sharp peak appears at 2θ = 7.9° with a broad peak observed with a 2θ = 20.0° maximum, as reported previously [16,17]. The sharp peak can be assigned to triple-helix molecular chains with a separation distance of 1.12 nm, as calculated using Bragg’s law. For the broad peak (disordered collagen chains), the chain distance was reduced up to 0.44 nm. This implies that the collagen contains ordered triple helix chains with a large chain separation as well as disordered collagen chains with smaller distances.

#### 2.3.6. Collagen Hydrolysis and Ultrafiltration

Collagen fibers were solubilized from *I. badionotus* body wall with pepsin, salted out, recovered by centrifugation, dialyzed and lyophilized. The fibrils were then digested with papain. The digestate was filtered and centrifuged. The degree of hydrolysis (DH) of the soluble collagen peptide solution was 79%.

The soluble collagen supernatant was passed through a 1 kDa cut-off membrane to separate it into collagen <1 kDa and >1 kDa fractions. The latter was then passed through a 3 kDa membrane to produce collagen 1–3 kDa and >3 kDa fractions.

In kit assays, the IC_50_ values for ACE and DPP4 inhibition for the collagen 1–3 kDa fraction were 270 µg/mL (1.85 U/mg) and 100 µg/mL (5.0 U/mg), respectively.

In the initial study, the collagen 1–3 kDa fraction was found to be the most potent of the ultrafiltered fractions in promoting wound healing in vitro. After fractionation by flash chromatography (Figure 6), three peaks (1, 2, and 3) were recovered and freeze-dried for further analysis.

Before use in the scratch wound assay, the collagen 1–3 kDa and collagen 1–3 kDa Peak 2 fractions were tested for cytotoxicity by the standard MTT method. No toxicity was evident when HaCat cells were mixed with fractions at a concentration of 0.25, 0.125, 0.0625 and 0.0031 mg/mL. The concentration of 0.2 mg/mL was therefore used in the subsequent wound-healing assay.

#### 2.3.7. Wound-Healing Assay

The peptide fractions of *I. badionotus* collagen were tested for the potency of wound healing in a scratch assay model with human keratinocyte (HaCat) cells in culture medium containing 10% fetal calf serum (Figure 7). Collagen 1–3 kDa and collagen 1–3 kDa Peak 2 (Figure 6) both promoted wound healing (Figure 7a,b), while collagen <1 kDa and collagen >3 kDa (Figure 6) exhibited little or no activity. Collagen 1–3 kDa Peak 2 had the highest activity (Figure 7).

Similar wound closure trends were observed when these fractions were tested in another laboratory. However, closure rates were slower, possibly because of the much lower levels of fetal calf serum (1%) in the culture medium (Appendix A).

#### 2.3.8. Gene Expression Analysis

Relative expression of *IL1α*, *IL6*, *TGFβ1*, *DSG1*, *DSG3* and *S100A7* genes in human keratinocytes collected from the wound healing assays after 24 h culture with *I. badionotus* preparations varied according to the individual fractions (Figure 8). When wounds were treated with 1–3 kDa BW, relative expression levels of *IL1α*, *IL6*, and *TGFβ1* were significantly lowered (below one time-fold) compared to those in the control media, but *S100A7* was significantly elevated. Only *IL1α* and *DSG1* expression was reduced when wounds were treated with collagen 1–3 kDa. This fraction increased *S100A7* expression, but to a lesser extent than with 1–3 kDa BW. In contrast, treatment with collagen 1–3 kDa Peak 2 elevated expression of *IL1α* and *IL6* but did not affect *S100A7* or other genes.

#### 2.3.9. Antioxidant Activities of Collagen Fractions

Previous studies have suggested an association between the antioxidant capacity of collagen peptides and their potential to promote wound healing. So, these activities were determined using the ABTS and ORAC methods (Figure 9). Fractions (<1 kDa, 1–3 kDa and >3 kDa) of *I. badionotus* collagen were each active in both assays, but the Trolox equivalent levels were by far the highest with collagen 1–3 kDa. Collagen 1–3 kDa Peak 2, obtained by flash chromatography, also had potent antioxidant activities. However, the balance between the activities changed. The Trolox equivalent levels in the ORAC assay for collagen 1–3 kDa Peak 2 were significantly higher than in collagen 1–3 kDa, from which it was derived, while ABTS assay values were unaltered. Indeed, the ORAC-ABTS ratio for collagen 1–3 kDa Peak 2 resembled that for 1–3 kDa BW (Figure 1).

## 3. Discussion

The primary purposes of this research were to isolate and characterize the collagen of *I. badionotus* and evaluate whether peptide fractions derived from it could promote wound healing and closure in a scratch wound model based on keratinocytes (HaCaT cells) in culture [18,19]. Although this property has previously been reported for collagen peptides from several sea cucumber species [7,8,20,21,22], the present study is, as far as we can ascertain, the first to demonstrate that low-molecular-weight peptides of collagen from skin-free *I. badionotus* body wall promote wound healing and closure in vitro. The study also revealed that some non-Collagen peptides from digests of *I. badionotus* body wall may have similar abilities to promote wound healing in vitro.

Proximal analysis of *I. badionotus* harvested for the present study off the coast of Telchac Puerto, Yucatan, Mexico, was generally similar to that of *I. badionotus* collected off Dzilam de Bravo, Yucatan, Mexico [23], and off the Caribbean coast of Colombia at Santa Marta [24]. Its composition was also similar to the general ranges expected for other commercial-sea cucumber species captured worldwide [6,25,26,27,28,29,30,31].

Sea cucumbers are rich sources of bioactive factors, including peptides [native and proteolytic digestion products] that can promote wound healing in vitro and in vivo [7,8,20,32]. The levels and potency of these individual bioactive components vary significantly depending on the species, growth environment, season, body composition, and the handling or processing of captured or harvested animals [9,10,11].

While some bioactive properties of derivatives from *I. badionotus* from Yucatan have previously been evaluated in vitro and in vivo [14,15,33], their wound-healing capacity has not. The remedial efficacy of these derivatives was generally similar to that reported for other commercially important sea cucumber species [7,8,21]. As with other species, the wound-healing bioactivity of *I. badionotus* was mainly associated with a low-molecular-weight fraction of the body wall digest [7,20], specifically one of 1–3 kDa. Again, as with other species, the body wall fraction of *I. badionotus* had significant antioxidant, inhibitory and antibacterial activities [34,35,36]. The antibacterial properties for *I. badionotus* 1–3 kDa BW digests were similar to those reported for Holothuria leucospilota, and other species [37,38,39,40].

Collagen is the major individual protein in the sea cucumber body wall. However, content can vary between 20% and 70% of total protein depending on species, growth environment, and final body weight [41,42]. The collagen content of *I. badionotus* from Yucatan was approximately 16.5% of the dry body wall, and thus similar to levels reported in most commercially harvested species (Appendix A) [23,41,42].

The collagen isolated from *I. badionotus* body wall was type I, which mainly consists of multiple α1 and α2 subunits [43]. The SDS-PAGE profile was similar to that for collagen from *Stichopus horrens*, *Holothuria scabra*, and *H. leucospilota* [44]. The amino acid composition, SEM, UV spectrum, FTIR spectroscopy, and CD studies confirmed its substantive similarity—though not complete identity—to collagens from other sea cucumber and marine species [22,44,45,46,47,48]. The denaturation temperature (Td) of *I. badionotus* collagen [32.5 °C] was in the high range for sea cucumber collagens and thus similar to those from *S. horrens* [32.8 °C], *H. scabra* [32.3 °C], and *H. leucospilota* [34.6 °C] [44]. However, the Td was much higher than for collagens from *Stichopus monotuberculatus* [30.2 °C] [49] *Parastichopus californicus* [18.5 °C] and others [25,50]. The high Td value for *I. badionotus* collagen will, at least in part, be due to its elevated imino acid content (Hyp and Pro); these play a critical role in steadying the triple helix structure of collagen and enhancing its thermal stability [43,51]. Overall, *I. badionotus* collagen composition and organization were similar to that for other sea cucumber collagens, though with some unique structural and reactive features (Appendix A).

In the present study, the 1–3 kDa fraction of *I. badionotus* body wall (1–3 kDa BW), the collagen 1–3 kDa fraction, and the collagen 1–3 kDa Peak 2 fraction each promoted wound healing and closure in the scratch wound assay with keratinocyte (HaCat) cells in culture. These findings are consistent with reports on the wound-healing properties of other sea cucumber species derivatives [8,20,21,52]. In the present study, collagen 1–3 kDa Peak 2 was the most potent fraction (95% closure in 24 h), while collagen 1–3 kDa, from which it was derived, was less efficient (78% closure in 24 h). The 1–3 kDa BW fraction was comparable to collagen 1–3 kDa Peak 2 (90% closure in 24 h).

Restitution in vitro or in vivo involves a multiplicity of cellular responses and changes that allow keratinocytes on the borders of a wound in a cell layer to proliferate and migrate to fill and close the gap. These events include a transient reduction in cell-cell interactions and partial release, which allows cells to spread and migrate across the wound. These actions are followed by repolarization and maturation as the wound is covered and healing is completed [53,54]. Such processes are strictly regulated by growth factors, hormones and cytokines, derived locally or from cells underlying or associated with the disrupted layer [18,19,55,56,57,58,59,60]. Since the Scratch Wound assay is based on a cell monolayer, wound healing does not involve growth factors, hormones, cytokines or factors from underlying cells or tissue layers. Nevertheless, the Scratch Wound assay is an established in vitro procedure for first-stage evaluation of the wound-healing potential of external bioactive factors or cell-modulating procedures [18,54,55,56,61,62,63,64,65,66,67].

Interleukin 1α (IL1α), Interleukin 6 (IL6), Transforming growth factor beta (TGFβ), Desmoglein-1 (DSG1), Desmoglein-3 (DSG3), and S100 calcium-binding protein A7 or psoriasin (S100A7) play important roles in various aspects of wound healing in HaCat (keratinocyte) cells. Their expression levels vary over time in untreated wounds, depending on the healing stage and the key repair processes being carried out [18,56,58,60]. The expression and timing of these genes can be influenced by external conditions that alter the efficacy and integrity of the repair process.

Many studies have demonstrated that antioxidants from natural sources, including small peptides of marine collagens, facilitate wound healing in vitro and in vivo. This is probably due to their ability to moderate and quench excessive oxidative stresses that can occur during key stages of natural repair, which severely limit healing efficacy [68,69,70,71]. Data from the present study suggests a positive correlation between the antioxidant activities, in particular peroxyl radical neutralising actions [72,73], of the tested peptide preparations and their efficacy in wound healing. However, the marked differences in gene expression at 24 h, associated with wound healing mediated by each tested sea cucumber fraction, suggest that the antioxidant activity protected and facilitated repair, but was merely one of the factors involved.

The purified collagen 1–3 kDa Peak 2 peptide fraction, the most potent promoter of wound healing in vitro, was associated with slightly elevated expression of IL-1α and IL-6 at 24 h. High expression of these proinflammatory factor levels is expected and is important for initiating wound repair; their levels usually fall below or near control cell levels as wound closure or near-closure occurs [18,56,58,60,61,62,65]. The present finding with the highly purified collagen 1–3 kDa Peak 2 peptide fraction is thus unexpected and problematic. Whether this proinflammatory signal resolves appropriately at full wound closure or leads to a prolonged inflammatory state at later time points is a critical question for future studies.

The exact mode of action of collagen peptides in wound repair in keratinocytes remains unclear. However, wound healing is mediated through cellular receptors for growth factors, such as those for epidermal growth factor (EGF), transforming growth factor-α (TGFα), and keratinocyte growth factor (KGF) [66,67] and an array of immune-pathway and cytokine-initiating receptors [74,75,76]. In many cases, simultaneous activation of two or more of these receptors in combination leads to better or more effective wound repair than activation of single receptors alone [66,67]. Previously reported studies have shown that some collagen peptides can act as receptor agonists and potentially activate key cellular receptors and associated signaling pathways involved in cellular metabolism, proliferation, and wound healing [74,75,76,77]. Their actions are usually beneficial but can also be detrimental, and they are highly dependent on the exact peptide structure [74,75,76,77].

In the present study, sea cucumber 1–3 kDa body wall (1–3 kDa BW), collagen 1–3 kDa and collagen 1–3 kDa Peak 2 peptides all promoted wound healing in the Scratch wound assay over 24 h. However, the gene expression profile at 24 h suggests their overall actions in facilitating that repair differed according to their actual composition.

The highly purified collagen 1–3 kDa Peak 2 peptides effectively promoted wound healing in vitro but were associated with background proinflammatory responses at 24 h. In contrast, the collagen digest (collagen 1–3 kDa) preparation, from which collagen 1–3 kDa Peak 2 was derived, induced a slower rate of wound healing, but one in which the gene expression profile at 24 h—including elevated S100A7, a marker of antimicrobial defence and epithelial differentiation—had strong similarities to those observed during wound repair in HaCat cells treated with various short- or long-term non-direct repair-promoting measures [18,56,58,60,61,62,65].

The collagen 1–3 kDa fraction contains many more collagen-derived peptides than its highly purified counterpart. These additional peptides may influence, modulate, or counteract some of the actions of the most potent peptides in collagen 1–3 kDa Peak 2. They may also bind to and activate different receptors and cell-signaling pathways, facilitating a more coordinated and efficient wound repair.

Sea cucumber body wall 1–3 kDa BW preparation was almost as effective as the collagen 1–3 kDa Peak 2 in promoting wound healing. Furthermore, the gene expression profile it elicited was similar to that in HaCat cells when wound healing was induced by various short- or long-term non-targeted treatments [18,56,58,60,61,62,65]. Indeed, the profile was closest to that expected during unstimulated repair.

Sea cucumber whole-body wall contains many proteins unrelated to collagen, in I. badionotus up to 40%, as well as other bioactive factors that may facilitate wound repair [8,14,20,78]. Fragments of these proteins or small molecular weight bioactive factors may act in synergy with the collagen peptides in the 1–3 kDa BW fraction to better modulate/integrate wound healing. These aspects of integrated wound repair induced by the 1–3 kDa BW fraction of *I. badionotus* require further study.

Taken together, these results indicate that the three sea cucumber peptide preparations examined in this study all promoted wound healing in vitro but suggest they did so by acting on repair processes, at least in part, in different ways.

*I. badionotus* 1–3 kDa BW, collagen 1–3 kDa, and collagen 1–3 kDa Peak 2 peptide fractions significantly aided wound healing in a Scratch Assay model. However, studies in vivo are needed to confirm whether the in vitro properties of these preparations will transfer to an in vivo system. Furthermore, the mechanisms of action of low molecular weight collagen peptides and the possible involvement of other small compounds in modulating or amplifying their actions remain unclear and require further study.

## 4. Materials and Methods

### 4.1. Materials and Reagents

Porcine stomach mucosa pepsin (EC3.4.23.1), dialysis membrane (14 kDa MWCO), and calf skin type I collagen standard solutions were purchased from Sigma-Aldrich (St. Louis, MO, USA); *Carica papaya* papain (30,000 U/mg; E.C. 3.4.22.2) and ultrafiltration membranes (1 and 3 kDa MWCO) from Merck Corporation (Burlington, MA, USA), and solvents for amino acid analysis were HPLC-grade from T.J. Baker Chemicals (Baker, PA, USA). All other chemicals were analytical grade.

### 4.2. Collection and Processing of Sea Cucumber

Wild sea cucumber *I. badionotus* was collected manually from the seafloor by scuba diving (approximately 10 metres depth). They were collected off the coast of Telchac Puerto, Yucatan, Mexico, under collection permit PPF/DGOPA-0126/15 issued by the National Commission for Fisheries and Aquaculture (CONAPESCA) in 2015 and renewed annually to date. All handling and processing of captured animals was carried out exactly as previously [54,55].

### 4.3. Proximate Composition

In the laboratory, the samples were partially freeze-dried (10 h), the skin removed, the body wall cut into squares and dried by lyophilization. The moisture and ash contents of samples of whole-body wall were determined following standard AOAC methods [15,33]; a portion of tissue was dried in an oven at 105 °C for 24 h to measure moisture content, and another was reduced in a muffle furnace at 550 °C for 6 h to measure ash content. Lipid content was estimated by the Folch method [15,33]. Nitrogen content was measured with an elemental analyzer (FLASH 2000, ThermoScientific, Waltham, MA, USA), and protein content was calculated by multiplying by the factor 6.25.

### 4.4. Skin-Free Body Wall Digest

Twenty grams of dry, skin-free body wall was washed twice with sterile cold water and incubated overnight in a water bath at 60 °C with 200 mL sodium acetate pH 6 buffer (50 mM sodium acetate, 50 mM L-cysteine and 5 mM EDTA) containing 2 g papain. The enzyme was deactivated by heating the liquor at 80 °C for 10 min and then cooled. The resulting liquor was first filtered through glass wool and then centrifuged at 10,000× *g*. The supernatant was recovered and sequentially ultrafiltered using an Amicon^®^ Stirred Cell (Merck) and ultrafiltration membranes with 1 kDa and 3 kDa cut-offs. The liquor was first passed through a 1 kDa cut-off membrane to obtain a >1 kDa retentate fraction and a <1 kDa permeate fraction. The >1 kDa retentate was then passed through a 3 kDa cut-off membrane to obtain 1–3 kDa and >3 kDa fractions [<1 kDa BW, 1–3 kDa BW and >3 kDa BW].

### 4.5. Pepsin-Soluble Collagen Extraction

Collagen from the skin-free body wall was extracted according to the method of Liu et al. [50], with slight modifications. All processing was done at 4 °C.

Fifty grams of dry, skin-free body wall were washed in one liter of cold deionized water under gentle agitation for 1 h (changed at 30 min). The water was replaced by the same volume of an ethylenediamine-tetra-acetic (EDTA) solution [4 mM in Tris-HCl (0.1 M, pH 8.0)] and stirred continuously (500 rpm at 4 °C) for 48 h with two changes per day. The remaining tissue pieces were recovered by sieving through cheesecloth and washing with cold deionized water until the effluent pH reached approximately neutral. The resulting precipitate was dispersed in 20 volumes (*w*/*v*) NaOH (0.1 M) and stirred for two days (500 rpm at 4 °C) to dissolve the non-collagenous components. The collagen fibers were recovered by gentle sieving and washing several times with cold deionized water until reaching a pH near neutral. They were dispersed in 0.5 M acetic acid containing 2% (*w*/*w*) pepsin and incubated for 72 h. The solubilized collagen fibers were salted out from the solution by adding NaCl to make it 1.2 M with respect to the salt and leaving the suspension for two days at 4 °C, followed by centrifugation at 15,000× *g* for 15 min. The resultant pellets were resuspended in acetic acid solution (0.1 M) and dialyzed (12,000–14,000 kDa membrane) in Na_2_HPO_4_ phosphate buffer (0.02 M, pH 8.0) for 48 h, with a solution change every 12 h. The purified collagen was freeze-dried and stored at 4 °C until use.

### 4.6. Collagen Characterization

#### 4.6.1. Electrophoresis

Sodium dodecyl sulfate-polyacrylamide gel electrophoresis (SDS-PAGE) analysis was performed according to the Laemmli method [79,80]. The lyophilized collagen was dissolved in 0.1 M acetic acid (final protein concentration, 4 mg mL^−1^) and then mixed at a 1:2 (*v*/*v*) ratio with the sample buffer (0.5 M Tris-HCl, pH 6.8, containing 5% SDS, 20% glycerol, 5% β-ME and 0.2% bromophenol blue). The mixed solution was incubated at 95 °C for 5 min and then subjected to electrophoresis (7.5% separator and 4% stacking gels) using a Mini-PROTEAN 3 Cell (Bio-Rad Laboratories, Hercules, CA, USA). After electrophoresis, samples were viewed by staining with Coomassie brilliant blue and destained in 40% methanol and 10% acetic acid. A molecular weight protein marker (Precision Plus Protein Standards All Blue, Bio-Rad) was used to estimate collagen molecular weight.

#### 4.6.2. Amino Acid Analysis

Lyophilized collagen was analyzed for amino acid composition according to the PicoTag procedure (Waters, Milford, MA, USA). Samples were hydrolyzed with 6 N HCl with 0.1% phenol and incubated in a nitrogen atmosphere at 110 °C for 22 h. After hydrolysis, samples and standards were derivatized with phenyl isothiocyanate (PITC) reagent and reconstituted in a sodium phosphate buffer (5 mM, pH 7.4) containing 5% (*v*/*v*) acetonitrile. The derivatives were analyzed by reverse phase chromatography (RP-UHPLC) in an Ultimate 3000 HPLC system (ThermoScientific, Waltham, MA, USA). Amino acid quantification was done using a standard amino acid mixture as a reference and expressed as the number of residues per 1000 total residues.

#### 4.6.3. X-Ray Diffraction (XRD) and Circular Dichroism (CD)

Collagen crystal structures were studied using an X-ray diffraction instrument (Bruker D8 Advance DaVinci, DE; Bruker Mexicana, Mexico City, Mexico) under these conditions: Cu Ka source, 40 kV tube voltage, 40 mA tube current, 3–60° scanning range (2θ), and 0.02°/s scanning speed. Secondary structure preservation of the collagen was assessed by Far-UV CD using a Chirascan spectropolarimeter (Applied Photophysics Ltd., Leatherhead, UK). The lyophilized collagen was dissolved in 0.1 M acetic acid at 0.1 mg mL^−1^ and continuously stirred at 4 °C for 24 h. Collagen solutions were placed in a quartz cell with a 0.1 cm path length. CD spectra were recorded in a 185–260 nm range at 4 °C, a 50 nm/min scan speed, and a 0.1 nm interval. Denaturation temperature (Td) was calculated from the ellipticity value at a fixed wavelength of 222 nm in the 4–65 °C range at a 1 °C/min rate.

#### 4.6.4. UV–Vis and Fourier Transform Infrared (FTIR) Spectra

The collagen UV spectra were obtained using a Multiskan GO spectrophotometer (Thermo Fisher Scientific, Waltham, MA, USA). Lyophilized collagen was dissolved in acetic acid at a 0.1 mg mL^−1^ concentration and continuously stirred at 4 °C for 12 h. The sample solution was placed in a quartz cell with a 10 mm path length. The spectra were recorded at 25 °C at a 1 nm interval in the 200–400 nm range. FITR spectra were recorded with a Nicolet iS5 (Thermo Scientific, México, Mexico) spectrophotometer using the ATR methodology. Lyophilized collagen was clamped onto the zinc selenide diamond crystal, and its spectra were recorded with a spectrometer at 2 cm^−1^ resolution in the 400–4000 cm^−1^ range. The baseline was set with 0.1 M acetic acid.

### 4.7. Collagen Hydrolysate Preparation

Hydrolysis of collagen was done with papain, according to Chotphruethipong et al. [81] and Vieira and Murao [82]. Ten grams of collagen were suspended in 100 mL acetate buffer (pH 6) and digested with papain (enzyme—substrate ratio, 1:10, *w*/*w*) for 12 h at 60 °C. The solution was heated at 80 °C for 5 min to inactivate the enzymatic reaction, filtered through glass wool and centrifuged at 5000× *g* for 30 min at 4 °C to obtain a clear supernatant. The resulting soluble collagen peptide solution was freeze-dried. Its degree of hydrolysis (DH) was measured by spectrophotometry following Nielsen et al. [83], using NAC (*n*-Acetyl-L-cysteine) as a thiol reagent, as described by Spellman et al. [84]; the DH was 79%.

#### 4.7.1. Ultrafiltration

Fractionation of the soluble collagen peptide solution was conducted as per Section 4.4, resulting in three fractions: collagen <1 kDa, collagen 1–3 kDa, and collagen >3 kDa.

#### 4.7.2. Flash Chromatography of Collagen 1–3 kDa

The 1–3 kDa isolate was reconstituted in deionized water (40 mg mL^−1^), loaded onto a SNAP Ultra C18 column (Biotage^®^ HP-Sphere, Charlotte, NC, USA) and stepwise eluted with water: acetonitrile (gradient of 0% to 40% acetonitrile), at a 25 mL/min flow rate, and 215 and 280 nm wavelengths. Fractions (5 mL) were collected, pooled in accordance with the graphic profile and lyophilized. Data was recovered from the printout (Appendix A) using Plotdigitizer (https://plotdigitizer.com, accessed on 22 March 2025) and printed out using GraphPad Prism (https://www.graphpad.com, accessed on 14 February 2025).

#### 4.7.3. Antioxidant Activity

##### ABTS+ Scavenging Activity

Scavenging antioxidant activity was measured by the 2,2-azinobis [3-ethylbenzothiazoline-6-sulfonic acid] (ABTS+) method [85]. Briefly, the radical ABTS+ stock solution was generated by mixing (*v*/*v*) 7 mM ABTS and 2.45 mM potassium persulfate in the dark at room temperature for approximately 16 h. The ABTS+ working solution was produced by diluting the stock solution in 5 mM sodium phosphate buffer (pH 7.4) at 734 nm absorbance and a 0.70 ± 0.02 range. The working solution (180 μL) was added to 20 μL peptide fraction sample or the standard 6-hydroxy-2,5,7,8-tetramethylchroman-2-carboxylic acid; (Trolox) in PBS. Absorbance was measured after 6 min in a spectrophotometer (Multiskan Go, ThermoScientific, Waltham, MA, USA). Samples were analyzed in three replicates. Percentage inhibition was calculated and plotted as a function of the amount of antioxidants (mg) or standard (µmol Trolox), and Trolox equivalent antioxidant capacity values were calculated.

##### Oxygen Radical Absorbance Capacity (ORAC)

The ORAC assay was conducted per Garret et al. [86]. Samples and standards were prepared in PBS (1 mg mL^−1^, double diluted and 20 µL of each dilution was placed in the wells of a 96-well plate. Two hundred microliters of fluorescein (3, 6-dihydroxyspiro [isobenzofuran-1[3H],9[9H]-xanthen]-3-one) [FL] stock (30 nmol L^−1^) and 75 µL 2, 2-azobis (2- amidinopropane) dihydrochloride [AAPH]) stock (12 mmol L^−1^) were added to each well. A standard curve was produced using an antioxidant calibrator (Trolox, 0–5 nmol L^−1^). The reaction was conducted at 37 °C, and fluorescence was recorded every 3.5 min for 31 cycles in an Apliskan plate reader (ThermoScientific, Waltham, MA, USA) at 485 nm excitation and 520 nm emission. Three replicates were done for each sample and standard. ORAC values were expressed as µmol Trolox equivalents (TE)/mg sample.

##### Further Analyses

Cytotoxicity of the 1–3 kDa BW, collagen 1–3 kDa, and collagen 1–3 kDa Peak 2 preparations was evaluated by the thiazolyl blue tetrazolium bromide (MTT) assay with keratinocyte (HaCat) cells, according to Mani & Swargiary [87]. The concentrations used in the assay were 0.5, 0.25, 0.125, 0.0625, and 0.0031 mg/mL. Angiotensin I-converting enzyme (ACE) and dipeptidyl peptidase 4 inhibitory activities were quantified using assay kits (C50002 and Mak088, Sigma-Aldrich, St. Louis, MO, USA). Both were done in triplicate according to the corresponding manufacturer’s protocol.

The minimum inhibitory concentrations of *I. badionotus* peptide preparations (IBP) were determined according to a microplate assay procedure [88]. IBP stock solution (8 mg/mL) was prepared in Mueller–Hinton medium, serial dilutions were prepared to give working stocks of 4 mg/mL, 2 mg/mL, 1 mg/mL, 0.5mg/mL, 0.25mg/mL, and 0.125mg/mL, and 100 µL of each was added to wells in a microplate.

Overnight cultures of *Escherichia coli* ATCC 25922, *Pseudomonas aeruginosa* ATCC 13637, *Pseudomonas aeruginosa* 64D (MDR), *Staphylococcus aureus* ATCC 25725, *Staphylococcus aureus* MR (Methicillin resistant), *Enterococcus faecalis* ATCC 51299 (Vancomycin resistant), and *Enterococcus faecium* 683D (Vancomycin resistant) were centrifuged, washed, and resuspended in 0.85% NaCl to a final concentration of ~1 × 10^8^ CFU/mL. Five µL (equivalent to ~5 × 10^5^ CFU) were then added to each well, and the microplates were then incubated at 37 °C for 18 h. The MIC was defined as the lowest antimicrobial compound concentration that prevented the growth of bacteria and was evaluated in three replicate assays.

To determine whether the observed inhibition was bacteriostatic or bactericidal, 10 µL from wells with full inhibition (4 mg/mL IBP over 18 h) were plated on Blood agar or Luria Bertani agar and incubated for a further 18 h at 37 °C. Growth on the agar indicated the IBPs were bacteriostatic, while the absence of any growth demonstrated they were bactericidal.

### 4.8. Wound Healing In Vitro

#### 4.8.1. Scratch Wound Healing Assay

Primary stocks of immortalized human keratinocytes (HaCat) cell line provided by Dr. Oscar Medina Contreras (Hospital Infantil de Mexico “Federico Gómez”, Mexico City) were cultured in DMEM/Ham’s F12 medium supplemented with bicarbonate, an antibiotic-antimycotic, and 10% FBS (GIBCO), at pH 7.2 [61,89]. Cells were seeded on treated 75 cm^2^ culture plates (NEST Scientific Inc., Rahway, NJ, USA) and incubated at 37 °C in 5% CO_2_. The culture medium was changed at three-day intervals.

The keratinocytes were harvested when 80% confluent. The medium was removed, the cell layer washed with PBS (pH 7.2), 3 mL trypsin-EDTA [Trypsin 0.05%—EDTA 0.53 mM] added to the plate, and the plate incubated at 37 °C for 4 min to detach the cells. Five mL culture medium was added to the plate, the total volume recovered and placed in a 15 mL centrifuge tube. After centrifugation at 1500 rpm for 5 min, the supernatant was removed, and 1 mL culture medium was added. The tube was mixed gently to disperse the cells, and counts were done using a manual hemocytometer. The cell preparation was then diluted with medium to a 160,000 cells/mL concentration.

For the assay, 300 µL medium containing 50,000 keratinocytes was added to each well of a 48-well plate (NEST Scientific Inc., Rahway, NJ, USA), and the plate was incubated at 37 °C in 5% CO_2_. Cell growth was monitored until 80% confluence was attained. The culture medium was then removed from each well, and the adherent cells were washed twice with 500 µL fresh, sterile PBS 7.2. During the second wash, a scratch wound was made in the cell layer using a one-ml pipette (blue) tip. The PBS pH 7.2 was removed, and a third wash was done.

Pre-prepared collagen peptide fractions (60 µg in 0.3 mL) or control medium (0.3 mL) were added to the appropriate wells [six replicates each], and the plates were incubated at 37 °C and 5% CO_2_. The healing process was monitored by taking images at 0 and 24 h post-wound induction via an inverted, phase-contrast microscope (10×). Wound healing rate was assessed using the ImageJ software V 1.54k (Gebäck, Zurich, Switzerland). Means and standard deviations were calculated, and statistical significance was evaluated by multiple comparisons and post-hoc significance tests using the GraphPad Prism.

#### 4.8.2. Gene Expression

Expression of several genes involved in the healing process was quantified (Table 2), using the β-actin gene as a housekeeping gene.

RNA extraction from human keratinocyte (HaCat) cells was done using a commercial Animal Tissue RNA Purification kit (Norgen Biotek Corp©, Thorold, ON, Canada), following the manufacturer’s protocol. Extracted RNA concentration and purity were assessed using a Thermo Scientific NanoDrop™ 2000c spectrophotometer (Thermo Fisher Scientific, Waltham, MA, USA). The extracted RNA samples were standardized to a 100 ng/µL final concentration, from which the cDNA of all samples was prepared using the commercial High-Capacity cDNA Reverse Transcription Kit (Thermo Fisher Scientific™), following the manufacturer’s protocol. Primer calibration curves were generated using five serial dilutions at a 1:5 dilution factor.

Quantification of gene expression was done by real-time quantitative PCR (qPCR) on a Rotor-Gene Q thermal cycler (QIAGEN^®^, Manchester, UK), using the commercial Maxima SYBR Green/ROX qPCR Master Mix/2X kit (Thermo Fisher Scientific™). Reaction mixes were standardized into a final volume of 15 μL containing 7.5 μL Maxima SYBR Green 2X reagent (QIAGEN), 1 µL diluted cDNA (5 µL cDNA + 20 µL water), 0.5 µL of each primer (5 μM), and 5.5 µL nuclease-free water. Amplification thermal cycling conditions were: initial denaturation at 95 °C for 10 min; 35 cycles of denaturation at 95 °C for 10 s; an extension at 60 °C for 45 s. Amplification specificity was confirmed by analyzing the dissociation curve of the formed products. The results were processed with the Rotor-Gene software version 6.1.

Evaluation of gene expression was done by evaluating three replicates of each sample from the 24 h scratch wound assay. The quantifications were based on the threshold cycle (Ct) value and the Ct values for each replicate. Changes in relative gene expression were calculated using the delta delta Ct (∆∆Ct) relative quantification method [91,92,93,94].

### 4.9. Statistical Analysis

GraphPad Prism (https://www.graphpad.com/) was used to conduct the statistical analysis of data from each of three independent experiments. All data were expressed as mean ± SD and analysed by *t*-test, or by one-way analysis of variance (ANOVA) and Tukey’s post hoc test or the Bonferroni post hoc test for multiple comparisons if one-way ANOVA tests indicated statistical significance, which was set at ≤0.05.

## 5. Conclusions

The body wall of *I. badionotus* from the Yucatan Peninsula contained collagen levels comparable to those in other commercially-harvested sea cucumber species. Collagen composition and structure were similar to, but not identical to, other sea cucumber collagens. Ultrafiltered digests (collagen 1–3 kDa) of the collagen and further purified samples from the collagen 1–3 kDa preparation (collagen 1–3 kDa Peak 2), as well as a 1–3 kDa digest of whole-body wall (1–3 kDa BW), had potent antioxidant activities, in particular, peroxyl radical neutralizing actions. They promoted rapid wound healing in a scratch assay using keratinocyte (HaCat) cells. However, expression of genes associated with wound repair differed significantly between cells treated with preparations of 1–3 kDa BW, collagen 1–3 kDa, or collagen 1–3 kDa Peak 2. Thus, low-molecular-weight peptides of *I. badionotus* collagen greatly aided wound healing in a monolayer model system. However, studies in vivo are needed to confirm that these healing properties will transfer to an in vivo system. Furthermore, the mechanisms of action of low molecular weight collagen peptides and the possible involvement of other small compounds in modulating or amplifying their actions remain unclear and require further study.

## Figures and Tables

**Figure 1 marinedrugs-23-00411-f001:**
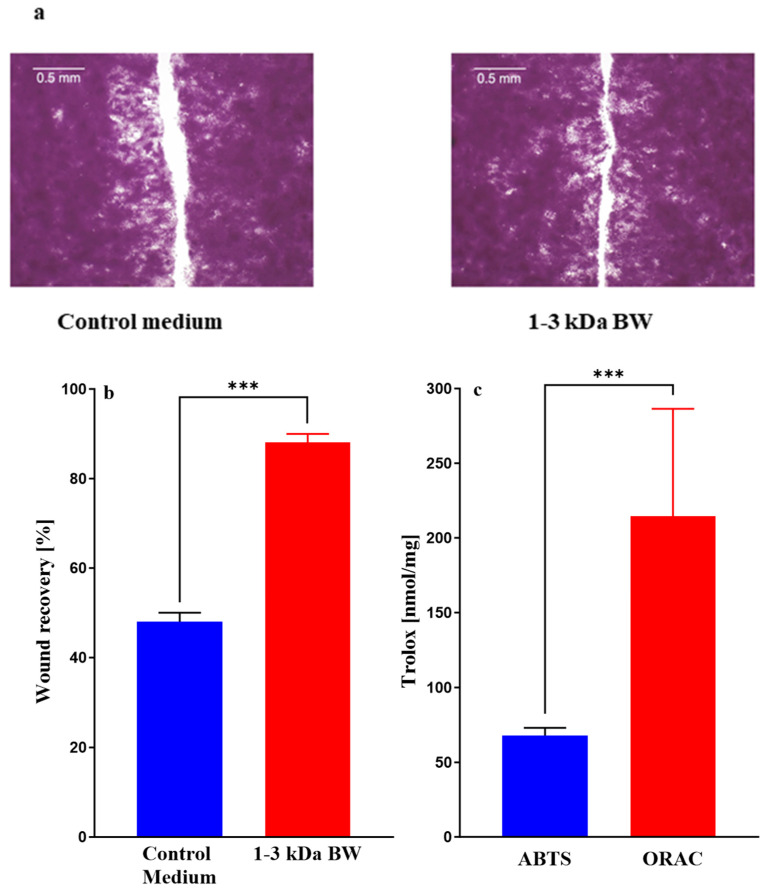
Images (**a**) and evaluations (**b**) of wound healing in a scratch assay based on human keratinocytes cultured for 24 h in media containing 10% fetal calf serum with or without ultrafiltrate fractions of digested body wall from *I. badionotus* (1–3 kDa BW) and the constituent antioxidant activities (ABTS and ORAC) for 1–3 kDa BW (**c**). For each data set, *n* = 3 and *** indicates a significant difference (*p* ≤ 0.001).

**Figure 2 marinedrugs-23-00411-f002:**
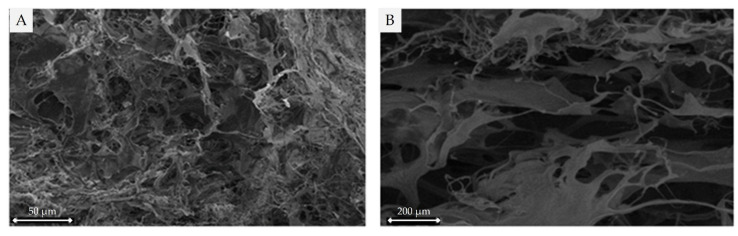
Scanning electron microscope (SEM) images of pepsin-soluble collagen from *I. badionotus* by secondary electrons (**A**) and by retro-disperse electrons (**B**).

**Figure 3 marinedrugs-23-00411-f003:**
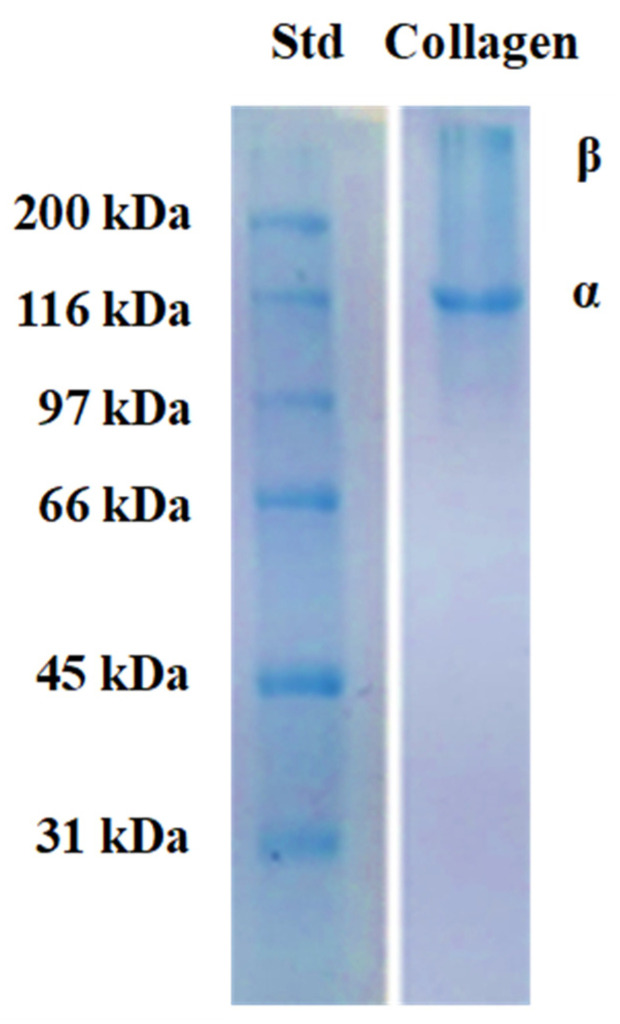
SDS-PAGE electrophoresis in reduced conditions of collagen from *I. badionotus* body wall. (Std: SDS-PAGE molecular weight standards broad range (Bio-Rad, Mexico City, Mexico) 6.5 kDa to 200 kDa).

**Figure 4 marinedrugs-23-00411-f004:**
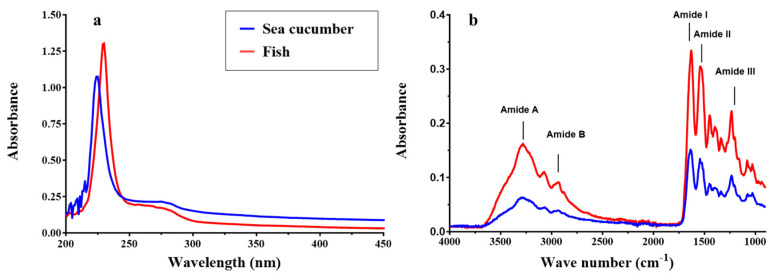
UV–visible spectra (**a**) and Fourier transform infrared spectroscopy (FTIR) (**b**) of collagen from the skin-free body wall of *I. badionotus* collagen and from the fish *Totoaba macdonaldi*.

**Figure 5 marinedrugs-23-00411-f005:**
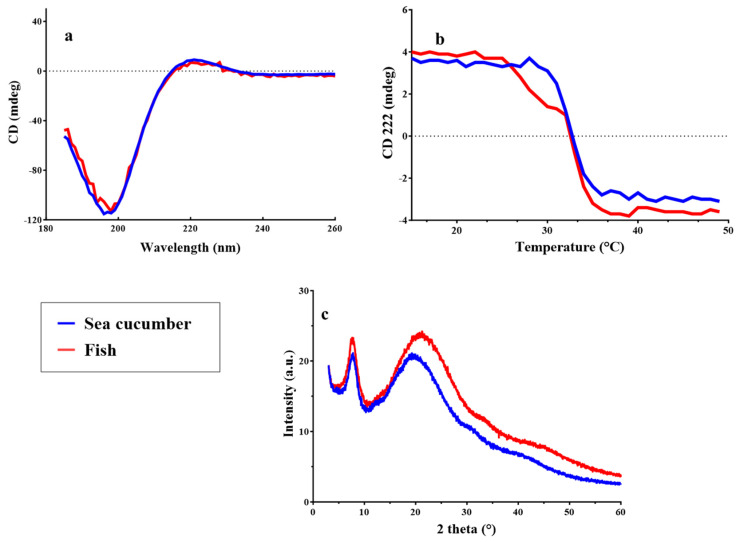
Far-UV circular dichroism spectra (**a**), temperature-induced unfolding followed at 222 nm (**b**) and X-ray diffraction (XRD) (**c**) of collagen from *I. badionotus* and from the fish *Totoaba macdonaldi*.

**Figure 6 marinedrugs-23-00411-f006:**
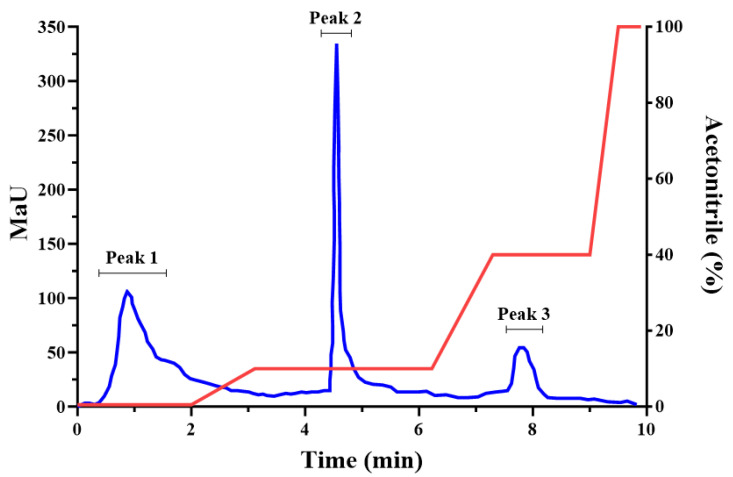
Graphic representation of separation of the *I. badionotus* collagen 1–3 kDa fraction by flash chromatography. Data was acquired from the original printout (Appendix A) with Plotdigitizer and drawn with GraphPad Prism 10.3.0.

**Figure 7 marinedrugs-23-00411-f007:**
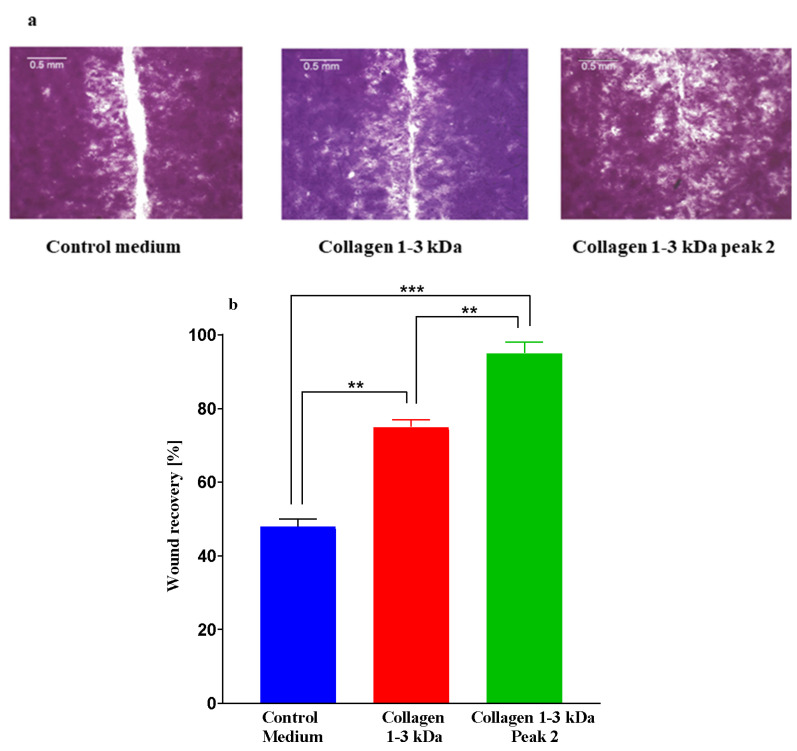
Images (**a**) and evaluations (**b**) of wound healing in a scratch assay based on human keratinocytes cultured for 24 h in media containing 10% fetal calf serum with or without ultrafiltrate fractions of digested collagen from *I. badionotus* (collagen 1–3 kDa) or a further purified preparation (collagen 1–3 kDa Peak 2). *n* = 3 and marked values differ significantly (** *p* ≤ 0.01; *** *p* ≤ 0.005).

**Figure 8 marinedrugs-23-00411-f008:**
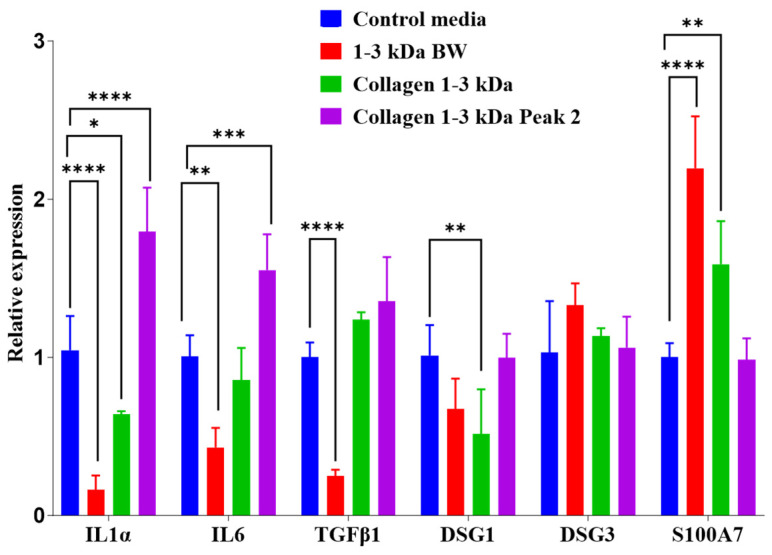
Expression of *IL1α*, *IL6*, *TGFβ1*, *DSG1*, *DSG3* and *S100A7* in human keratinocytes recovered from the wound-healing assay after culture [10% fetal calf serum] for 24 h with 1–3 kDa BW, collagen 1–3 kDa or collagen 1–3 kDa Peak 2. *n* = 3 and * *p* ≤ 0.05, ** *p* ≤ 0.01, *** *p* ≤ 0.001, **** *p* ≤ 0.0001.

**Figure 9 marinedrugs-23-00411-f009:**
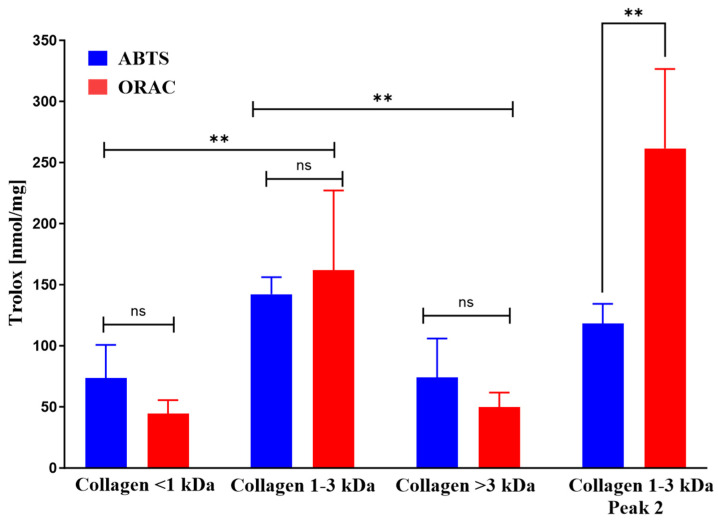
The constituent antioxidant activities (ABTS and ORAC) of ultrafiltrate fractions of digested collagen from *I. badionotus* or collagen 1–3 kDa Peak 2. *n* = 3 and values with ** differ significantly (*p* ≤ 0.01). ns, not significant.

**Table 1 marinedrugs-23-00411-t001:** Amino acid composition of pepsin-soluble ollagen from skin-free body wall of *I. badionotus*.

Amino Acid	Residues *	Amino Acid	Residues *
Aspartic acid	93	Tyrosine	6
Glutamine	157	Valine	20
Hydroxyproline	97	Methionine	5
Serine	21	Cysteine	1
Glycine	289	Isoleucine	10
Histidine	3	Leucine	15
Arginine	54	Hydroxylysine	5
Threonine	23	Phenylalanine	9
Alanine	110	Lysine	5
Proline	75	Imino acids **	172

* Residues/1000 residues. ** Proline + Hydroxyproline.

**Table 2 marinedrugs-23-00411-t002:** Primer sequences used for real-time quantitative PCR.

Target Gene	Forward 5′—3′ Reverse 3′—5′	Reference
*ILlα*	F: CGCCAATGACTCAGAGGAAGA R: AGGGCGTCATTCAGGATGAA	Wiegand et al., 2021 [56]
*IL6*	F: AGACAGCCACTCACCTCTTCAG R: TTCTGCCAGTGCCTCTTTGCTG	NM_000600.5
*TGFβ1*	F: GAGCCCTGGATACCAACTATT R: AGGACCTTGCTGTACTGTGTG	Wallace et al., 2023 [90]
*DDSG1*	F: TCCCCACATTTCGGCACTAC R: GCCCAGAGGATCGAGAATAGG	Wiegand et al., 2021 [56]
*DSG3*	F: GTCAGAACAATCGGTGTGAGATG R: TGCGGCCTGCCATACCT	Wiegand et al., 2021 [56]
*SI00A7*	F: GTCCAAACACACACATCTCACT R: TCATCATCGTCAGCAGGCTT	Wiegand et al., 2021 [56]
*β-actin*	F: GATCATTGCTCCTCCTGAGC R: GTCATAGTCCGCCTAGAAGCAT	NM_001101.5

## Data Availability

The original contributions presented in this study are included in the article/Appendix A. Further inquiries can be directed to the corresponding author.

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
