# Peer review of "Sea Cucumber (Isostichopus badionotus): Bioactivity and Wound Healing Capacity In Vitro of Small Peptide Isolates from Digests of Whole-Body Wall or Purified Collagen"

_marinedrugs, 2025, doi:10.3390/md23110411_

Round 1

Reviewer 1 Report

Comments and Suggestions for Authors

Dear authors/editors:     

 It is a great honor and pleasure for me to be invited as the reviewer for this great work entitled “Sea Cucumber (Isostichopus badionotus): Bioactivity and wound healing capacity in vitro of small peptide isolates from digests of whole-body wall or purified collagen”. Leticia Olivera-Castillo and co-authors have delved into this topic presenting a thorough biochemical and functional evaluation of low–molecular weight peptides (1–3 kDa) derived from Isostichopus badionotus body wall and purified collagen, with emphasis on wound healing, antioxidant, antibacterial, and gene-expression effects in vitro. This study topic is interesting and important for marine-derived nutraceuticals and biomedical applications, attributing to their long-term efforts and contributions in this scientific field. Before the paper is ready for publication, I have a number of comments concerning this study:

Major Comments

  1. The study is the first to characterize I. badionotus collagen peptides for wound healing, expanding knowledge of marine bioactives. However, the novelty is partly incremental since similar findings exist for other sea cucumbers (Holothuria scabra, Stichopus horrens). Authors should sharpen the unique contribution of I. badionotus (e.g., its higher denaturation temperature, distinct amino acid composition, or stronger antibacterial profile).

2.The scratch wound assay is appropriate for preliminary screening, but in vitro keratinocyte assays alone are insufficient to claim wound-healing efficacy. The discussion should acknowledge the lack of in vivo validation (animal wound models, 3D skin equivalents). The gene expression analysis is intriguing, but the time frame (24 h) is too narrow. Wound repair is dynamic; additional timepoints (e.g., 6 h, 48 h) would provide stronger mechanistic insights.

3.The authors propose that different fractions act through distinct mechanisms (early inflammatory vs. barrier reinforcement). While plausible, the conclusions are highly speculative without direct pathway interrogation (e.g., signaling inhibitors, receptor binding assays).

4.The elevation of IL-1α and IL-6 by Collagen 1–3 kDa Peak 2 is interpreted as beneficial, yet prolonged inflammation can impair healing. This should be discussed more critically.

5.The data show bactericidal effects against Gram-negative bacteria but only bacteriostatic activity against Gram-positive bacteria at 4 mg/mL. This concentration is relatively high and may not be physiologically relevant. The authors should address whether the antibacterial effect is clinically meaningful or simply a non-specific high-dose effect.

6.Figures should consistently display error bars and replicate numbers. For example, Figures 7–9 report significant differences but do not clearly indicate n-values. Also, the use of multiple statistical comparisons increases false-positive risk. Was correction for multiple testing applied (e.g., Bonferroni or FDR)?

7.The conclusion suggests that collagen peptides could be developed as wound-healing nutraceuticals or biomedical agents. This statement is premature without animal validation, bioavailability studies, or toxicology assessments. Authors should temper claims accordingly.

Minor Comments

1.The abstract is overly detailed and technical; simplify language for accessibility.

2.Replace phrases such as “potent promoter of wound healing” with more cautious wording like “significantly enhanced wound closure in vitro.”

3.Figure quality (microscopy, chromatograms) should be improved for readability.

4.Provide a summary table comparing collagen composition and denaturation temperatures across sea cucumber species (cited in Discussion).

5.Use consistent terms for fractions (“Collagen 1–3 kDa,” “Peak 2,” etc.). At times, labels are confusing.

6.Ensure all comparative statements (e.g., about denaturation temperatures, imino acid content, or antibacterial activity) are fully referenced.

  Thank you for giving me the opportunity to review this interesting article. Nonetheless, extensive editing of English language and style may be required.

Author Response

Reviewer 1

Dear authors/editors:    

 It is a great honor and pleasure for me to be invited as the reviewer for this great work entitled “Sea Cucumber (Isostichopus badionotus): Bioactivity and wound healing capacity in vitro of small peptide isolates from digests of whole-body wall or purified collagen”. Leticia Olivera-Castillo and co-authors have delved into this topic presenting a thorough biochemical and functional evaluation of low–molecular weight peptides (1–3 kDa) derived from Isostichopus badionotus body wall and purified collagen, with emphasis on wound healing, antioxidant, antibacterial, and gene-expression effects in vitro. This study topic is interesting and important for marine-derived nutraceuticals and biomedical applications, attributing to their long-term efforts and contributions in this scientific field. Before the paper is ready for publication, I have a number of comments concerning this study:

Major Comments

The study is the first to characterize I. badionotus collagen peptides for wound healing, expanding knowledge of marine bioactives. However, the novelty is partly incremental since similar findings exist for other sea cucumbers (Holothuria scabra, Stichopus horrens). Authors should sharpen the unique contribution of I. badionotus (e.g., its higher denaturation temperature, distinct amino acid composition, or stronger antibacterial profile).

A table comparing important parameters of collagen from I. badionotus and other species has been prepared and is included as supplemental Figure 1.

Species

Yield

(% DW)

α-chain (kDa)

Td

(°C)

Imino acids (%)

Gly (%)

Ala (%)

S. japonicas

26.6

135

35.3

19.2

32.5

9.8

H. Scabra

8.2

110-130

32.3

18.5

18.2

10.5

I. badionotus

16.5

131

32.5

17.2

28.9

11.0

H. nobilis

28.8

80-90

34.6

16.8

28.0

11.0

S. vastus

21.3

122

21.3

16.4

32.2

10.8

S. horrens

33

125

30.0

16.1

32.5

11.0

A. leucoprocta

44.0

110-130

25.4

16.0

30.0

9.0

H. arenicola

17.0

125

34.6

15.9

32.5

11.3

H. cinerascens

72.2

80-90

30.0

15.8

31.0

11.0

H. parva

7.0

130

32.5

15.8

27.0

9.1

H. leucospilota

27-30

133-166

34.6

15.8

43.8

2.2

S. monotuberculatus

2.6

137

30.2

15.1

32.0

9.2

P. californicus

20.0

138

17.9

14.2

32.5

11.2

A. mollis

10.0

116

46.7

8.8

28.6

15.9

References: [4][6][20][32][40][42-43][48] [95-105]

2.The scratch wound assay is appropriate for preliminary screening, but in vitro keratinocyte assays alone are insufficient to claim wound-healing efficacy. The discussion should acknowledge the lack of in vivo validation (animal wound models, 3D skin equivalents). The gene expression analysis is intriguing, but the time frame (24 h) is too narrow. Wound repair is dynamic; additional timepoints (e.g., 6 h, 48 h) would provide stronger mechanistic insights.

The text has been amended to emphasise that the data is in vitro and that follow up studies are necessary to confirm efficacy in vivo.

Ln 51-52. These findings need to be confirmed in vivo.

Ln 431.  I. badionotus 1-3 kDa BW, collagen 1-3 kDa, and collagen 1-3 kDa Peak 2 peptide fractions significantly aided wound healing in a Scratch Assay model. However, studies in vivo are needed to confirm the properties of these preparations will transfer to an in vivo system. Furthermore, the mechanisms of action of low molecular weight collagen peptides and the possible involvement of other small compounds in modulating or amplifying their actions remain unclear and require further study.

Ln 684. Thus, low-molecular-weight peptides of I. badionotus collagen greatly aided wound healing in a monolayer model system. However, studies in vivo are needed to confirm these properties will transfer to an in vivo model. Furthermore, the mechanisms of action of low molecular weight collagen peptides and the possible involvement of other small compounds in modulating or amplifying their actions remain unclear and require further study.

Time points for wound healing assay.

The wound healing procedure was monitored at 6, 12 and 24 h but only data at 24 h was fully analysed. At this timepoint, wound closure with cells treated with peptides was near but not complete and considered best to evaluate longer-term changes in cellular metabolism and gene expression during healing.

Ln 339- 436. The gene expression profiles and their implications for interpretating mechanisms of wound repair are more clearly and less speculatively dealt within the modified Discussion

3.The authors propose that different fractions act through distinct mechanisms (early inflammatory vs. barrier reinforcement). While plausible, the conclusions are highly speculative without direct pathway interrogation (e.g., signaling inhibitors, receptor binding

assays).

Ln 339-436. This aspect is based on reports in the literature on the potential modes of action of collagen peptides and indeed other peptides and was included as a possible explanation as to the significant differences in the actions of the three sea cucumber preparations evaluated in this study. A detailed examination of the mechanisms forms part of our ongoing research. This issue is more clearly dealt within the modified Discussion

4.The elevation of IL-1α and IL-6 by Collagen 1–3 kDa Peak 2 is interpreted as beneficial, yet prolonged inflammation can impair healing. This should be discussed more critically.

Ln 339-436. This issue is more critically discussed within the modified Discussion

5.The data show bactericidal effects against Gram-negative bacteria but only bacteriostatic activity against Gram-positive bacteria at 4 mg/mL. This concentration is relatively high and may not be physiologically relevant. The authors should address whether the antibacterial effect is clinically meaningful or simply a non-specific high-dose effect.

The text regarding the bacteriology data has been rewritten and expanded. The concentration of peptides necessary to cause inhibition does appear to be high, but it is near to the levels reported by others for Holothuria leucospilota, and other species (Ghanbari et al, 2012; Adibpour et al, 2014; Darya et al., 2020; Hossain et al, 2020) [refs. 36-39].

Material and Methods: Ln 590

The minimum inhibitory concentrations of I. badionotus peptide preparations (IBP)were determined according to a microplate assay procedure (KadeÅ™ábková et al. 2024) [88]. IBP stocks solution (8 mg/mL] were prepared in Mueller–Hinton medium, serial dilutions prepared to give working stocks of 4 mg/mL, 2 mg/mL, 1 mg/mL, 0.5mg/mL, 0.25mg/mL, and 0.125mg/mL and 100 µL of each working stock transferred added to wells in a microplate.

Overnight cultures of Escherichia coli ATCC 25922, Pseudomonas aeruginosa ATCC 13637, Pseudomonas aeruginosa 64D (MDR), Staphylococcus aureus ATCC 25725, Staphylococcus aureus MR (Methicillin resistant), Enterococcus faecalis ATCC 51299 (Vancomycin resistant) and Enterococcus faecium 683D (Vancomycin resistant) were centrifuged, washed, and resuspended in 0.85% NaCl to a final concentration of  ~1 x 108 CFU/mL. Five µL (equivalent to ~5 x 105 CFU) were then added to each well and the microplates were then incubated at 37 °C for 18 h. The MIC was defined as the lowest antimicrobial compound concentration that prevented growth of bacteria and was evaluated in three replicate assays.

To determine whether the observed inhibition was bacteriostatic or bactericidal, ten µL from wells with full inhibition (4 mg/mL IBP over 18 h) were plated on Blood agar or Luria Bertani agar and incubated for a further 18 h at 37 °C. Growth on the agars indicated the IBPs were bacteriostatic while the absence of any growth demonstrated they were bactericidal.

Ln 117. Results: The antibacterial capacity of the I. badionotus preparation (1-3 kDa BW) was tested against 5 x 105 CFU of Escherichia coli ATCC 25922, Pseudomonas aeruginosa ATCC 13637, Pseudomonas aeruginosa 64D (MDR), Staphylococcus aureus ATCC 25725, Staphylococcus aureus MR (Methicillin resistant), Enterococcus faecalis ATCC 51299 (Vancomycin resistant) and Enterococcus faecium 683D (vancomycin resistant). Following incubation for 18 h at 37 °C, no bacterial growth was observed with any of the tested bacterial species cultured in 4 mg/mL 1-3 kDa BW, variable growth was evident with 2 mg/mL 1-3 kDa BW, while expected growth occurred at 1-3 kDa BW concentrations of ≤ 1 mg/mL.

Ten µL of the bacterial suspensions that had been incubated for 18 h in 4 mg/mL 1-3kDa BW were then plated out on Blood agar or Luria Bertani agar and incubated at 37 °C for 18 h to evaluate viability. Significant growth was evident with Staphylococcus aureus ATCC 25725, Staphylococcus aureus MR (Methicillin-resistant), Enterococcus faecalis ATCC 51299 (Vancomycin-resistant) and Enterococcus faecium 683D (Vancomycin-resistant). In contrast, no growth was evident with Escherichia coli ATCC 25922, Pseudomonas aeruginosa ATCC 13637, or Pseudomonas aeruginosa 64D (MDR). These results indicate that, at a concentration of 4 mg/mL I. badionotus 1-3 kDa BW peptides were bacteriostatic against Gram-positive bacteria, but bactericidal to the Gram-negative bacteria tested.

Ln 316 Discussion: The antibacterial properties for I. badionotus 1-3 kDa BW digests were like those reported for Holothuria leucospilota, and other species [36-39].

6.Figures should consistently display error bars and replicate numbers. For example, Figures 7–9 report significant differences but do not clearly indicate n-values. Also, the use of multiple statistical comparisons increases false-positive risk. Was correction for multiple testing applied (e.g., Bonferroni or FDR)?

N-values added to all figures. Statistical analysis section expanded.

Ln 668. GraphPad Prism (https://www.graphpad.com/) was used to conduct the statistical analysis of data from each of three independent experiments. All data was expressed as mean ± SD and analysed by t-test, or by one-way analysis of variance (ANOVA) and Tukey’s post hoc test or the Bonferroni post hoc test for multiple comparisons if one-way ANOVA tests indicated statistical significance, which was set at ≤ 0.05.

7.The conclusion suggests that collagen peptides could be developed as wound-healing nutraceuticals or biomedical agents. This statement is premature without animal validation, bioavailability studies, or toxicology assessments. Authors should temper claims accordingly.

Text amended

Ln 51-52. These findings need to be confirmed in vivo.

Ln 431.  I. badionotus 1-3 kDa BW, collagen 1-3 kDa, and collagen 1-3 kDa Peak 2 peptide fractions significantly aided wound healing in a Scratch Assay model. However, studies in vivo are needed to confirm the properties of these preparations will transfer to an in vivo system. Furthermore, the mechanisms of action of low molecular weight collagen peptides and the possible involvement of other small compounds in modulating or amplifying their actions remain unclear and require further study.

Ln 684. Thus, low-molecular-weight peptides of I. badionotus collagen greatly aided wound healing in monolayer model system. However, studies in vivo are needed to confirm these properties will transfer to an in vivo model. Furthermore, the mechanisms of action of low molecular weight collagen peptides and the possible involvement of other small compounds in modulating or amplifying their actions remain unclear and require further study.

Minor Comments

1.The abstract is overly detailed and technical; simplify language for accessibility.

The abstract has been shortened and simplified.

2.Replace phrases such as “potent promoter of wound healing” with more cautious wording like “significantly enhanced wound closure in vitro.”

This phraseology has been removed and need for follow up in vivo studies emphasised.

3.Figure quality (microscopy, chromatograms) should be improved for readability.

Figure quality has been checked and enhance. Note that the original printout on which Figure 6 is based is given as Supplemental Figure 1.

4.Provide a summary table comparing collagen composition and denaturation temperatures across sea cucumber species (cited in Discussion).

A summary table of important collagen parameter across sea cucumber species has been prepared and fully referenced, Supplemental Table 1.

5.Use consistent terms for fractions (“Collagen 1–3 kDa,” “Peak 2,” etc.). At times, labels are confusing.

All labels have been checked

6.Ensure all comparative statements (e.g., about denaturation temperatures, imino acid content, or antibacterial activity) are fully referenced.

A summary table of important collagen parameter across sea cucumber species has been prepared and fully referenced, Supplemental Table 1.

  Thank you for giving me the opportunity to review this interesting article. Nonetheless, extensive editing of English language and style may be required.

The English has been checked by native English speakers.

Reviewer 2 Report

Comments and Suggestions for Authors

 The study asks whether low-molecular-weight peptides from the sea cucumber Isostichopus badionotus promote keratinocyte wound closure and how their actions relate to antioxidant capacity and antibacterial effects. The authors isolate collagen, generate ultrafiltrate peptide fractions (notably 1–3 kDa and a sub-fraction “Peak 2”), and evaluate activity in HaCaT scratch assays, antioxidant tests (ABTS/ORAC), limited antibacterial assays, and 24 h gene-expression profiling. In summary, the 1–3 kDa body-wall and collagen fractions accelerate wound closure in vitro; Peak 2 is the most potent. Antioxidant potencies differ across fractions and correlate with activity. Antibacterial testing at 4 mg/mL suggests bacteriostatic effects on Gram-positive and bactericidal effects on Gram-negative bacteria. Peak 2 elevates IL1α/IL6 at 24 h, while other fractions show different transcriptional signatures. Despite interesting, the manuscript must be revised before being considered for publication. 

Major issues

  1. Lines 239-241: The manuscript relies on size selection and a chromatographic “Peak 2” without mass spectrometry or sequence data. Figure 6 was reconstructed from a printout using PlotDigitizer, which underscores the need to show and deposit the original chromatogram and identify dominant peptides (LC-MS/MS).
  2. Conclusions about bacteriostatic vs bactericidal effects derive from a single high concentration (4 mg/mL) and post-incubation plating; no MIC/MBC series, replicates, or positive antibiotic controls are reported. Please perform CLSI-style MIC and MBC assays.
  3. Lines 636–638; 667–669; 259–262: Methods mention “multiple comparisons and post-hoc tests” but specific tests, assumptions, and exact n for each dataset are not stated; Figure 7 caption has an evident p-value asterisk error. Please specify tests (e.g., one-way ANOVA + Tukey), normality checks, n (biological vs technical), and correct captions. 
  4. The wound-healing assay appears to be conducted with limited replicates. While six wells are mentioned, statistical robustness and independent repeats are unclear. Were assays repeated in different batches?
  5. he manuscript claims this is the “first” demonstration of collagen-derived peptides from I. badionotus in wound healing. However, related studies on sea cucumber species exist. The novelty should be more carefully contextualized against prior work.
  6. The interpretation of gene expression is somewhat speculative. For example, upregulation of IL-6 and IL-1α is linked to positive healing, yet these are pro-inflammatory cytokines. Please provide stronger justification or reference suppor
  7. Lines 386-400. The discussion suggests peptide-receptor interactions but presents no experimental evidence (e.g., receptor binding assays).
  8. Lines 118–132, 602–609: The antibacterial results are promising but inconsistently reported. The text alternates between bacteriostatic and bactericidal terms without clarity. Please standardize and explain whether MIC or MBC values were determined.
  9. Statistical approaches are briefly mentioned (ANOVA/post-hoc), but exact tests, number of replicates, and assumptions (normality, variance homogeneity) are not explained. Please clarify.
  10. Some steps in sample handling and collagen extraction are insufficiently detailed. For example, what was the yield of collagen? How reproducible were extraction batches?  Minor issues
  11. Line 2–4: Title is long; consider shortening for readability while keeping the key terms “sea cucumber,” “collagen peptides,” and “wound healing.”

  12. Line 39: Duplicate “Correspondence” word should be corrected.

  13. Line 46–47: “like, but not identical to” is colloquial; replace with “similar, though distinct from.”

  14. Line 62–64: “remarkable capacity for regeneration” sounds non-scientific; suggest a more formal phrasing.

  15. Line 91–92: Collagen proportion (50%) should cite a specific reference or provide standard deviation/error.

  16. Line 103–105: The phrase “significant wounds remained” is unclear; suggest “wound closure was incomplete.”

  17. Line 125–126: Clarify whether “10 µL” plating was in duplicates or triplicates.

  18. Line 138–139: Methods list is long and redundant. Consider condensing or grouping by type.

  19. Line 171–174: Please clarify whether the absorption band at 258 nm is consistently reproducible.

  20. Line 199–202: The denaturation temperature (Td) is given without error margins. Please report mean ± SD.

  21. Line 229–231: “bacteriostatic against Gram-positive but bactericidal against Gram-negative” requires numerical evidence.

  22. Line 245: Cytotoxicity is described as absent, but higher doses were not reported. Please specify limits.

  23. Line 300–304: Comparisons to other studies could benefit from a concise table summarizing species, composition, and Td values.

  24. Line 344–345: “Almost as good as” sounds informal; replace with “comparable to.”

  25. Line 447–454: The animal collection process includes unnecessary detail (“plastic bags at depth”). This could be shortened.

  26. Several references (e.g., [16], [52]) are cited multiple times without clarity. Ensure consistency and update with most recent literature.

Author Response

Reviewer 2

Comments and Suggestions for Authors

 The study asks whether low-molecular-weight peptides from the sea cucumber Isostichopus badionotus promote keratinocyte wound closure and how their actions relate to antioxidant capacity and antibacterial effects. The authors isolate collagen, generate ultrafiltrate peptide fractions (notably 1–3 kDa and a sub-fraction “Peak 2”), and evaluate activity in HaCaT scratch assays, antioxidant tests (ABTS/ORAC), limited antibacterial assays, and 24 h gene-expression profiling. In summary, the 1–3 kDa body-wall and collagen fractions accelerate wound closure in vitro; Peak 2 is the most potent. Antioxidant potencies differ across fractions and correlate with activity. Antibacterial testing at 4 mg/mL suggests bacteriostatic effects on Gram-positive and bactericidal effects on Gram-negative bacteria. Peak 2 elevates IL1α/IL6 at 24 h, while other fractions show different transcriptional signatures. Despite interesting, the manuscript must be revised before being considered for publication. 

Major issues

  1. Lines 239-241: The manuscript relies on size selection and a chromatographic “Peak 2” without mass spectrometry or sequence data. Figure 6 was reconstructed from a printout using PlotDigitizer, which underscores the need to show and deposit the original chromatogram and identify dominant peptides (LC-MS/MS).

We fully agree that identification of the dominant wound-healing peptides of I. badionotus collagen 1-3 kDa peak 2 by LC-MS/MS is an important step, as done for collagens of other sea cucumber species. However, we have not done this to date because our data indicates the wound-healing promoted by peptides from I. badionotus, while in large part is due to collagen peptides is not due exclusively to the action of collagen-derived peptides. Rather, peptides from soluble proteins may contribute significantly. Also, comparisons between the efficacies of the collagen 1-3 kDa and collagen 1-3 kDa peak 2 preparations suggest that collagen peptides not included in collagen 1-3 kDa peak 2 may be important for efficient wound healing. So, identifying specific peptides from collagen 1-3 kDa peak 2 at this stage may be misleading and lead to the perception that the collagen 1-3 kDa peptides are the only ones that are critical. Before proceeding with LC-MS/MS identification of specific peptides, we intend to investigate the potential of peptides from the soluble protein component (at least forty percent of protein in the body wall) for potential to promote wound healing itself or additively/synergistically with collagen peptides. That information would greatly help in identification of the appropriate peptides to synthesise. Nonetheless, we consider it important at this stage to present our data, that indicates the low molecular weight peptides of collagen of I. badionotus, while promoters of wound healing, are not the only peptide factors with that property.

Ln 238. Figure 6 was acquired from the original printout with Plotdigitizer and drawn with GraphPad Prism. The original was included as Supplementary Figure 1 with the original text and flagged as such in the legend of Figure 6. The original will be Supplementary Figure 1 with the updated text.

  1. Conclusions about bacteriostatic vs bactericidal effects derive from a single high concentration (4 mg/mL) and post-incubation plating; no MIC/MBC series, replicates, or positive antibiotic controls are reported. Please perform CLSI-style MIC and MBC assays.
  1. Lines 118–132, 602–609: The antibacterial results are promising but inconsistently reported. The text alternates between bacteriostatic and bactericidal terms without clarity. Please standardize and explain whether MIC or MBC values were determined.
  2. Line 229–231: “bacteriostatic against Gram-positive but bactericidal against Gram-negative” requires numerical evidence

Expanded text

Material and Methods: Ln 590

The minimum inhibitory concentrations of I. badionotus peptide preparations (IBP)were determined according to a microplate assay procedure (KadeÅ™ábková et al. 2024) [88]. IBP stocks solution (8 mg/mL] were prepared in Mueller–Hinton medium, serial dilutions prepared to give working stocks of 4 mg/mL, 2 mg/mL, 1 mg/mL, 0.5mg/mL, 0.25mg/mL, and 0.125mg/mL and 100 µL of each working stock transferred added to wells in a microplate.

Overnight cultures of Escherichia coli ATCC 25922, Pseudomonas aeruginosa ATCC 13637, Pseudomonas aeruginosa 64D (MDR), Staphylococcus aureus ATCC 25725, Staphylococcus aureus MR (Methicillin resistant), Enterococcus faecalis ATCC 51299 (Vancomycin resistant) and Enterococcus faecium 683D (Vancomycin resistant) were centrifuged, washed, and resuspended in 0.85% NaCl to a final concentration of  ~1 x 108 CFU/mL. Five µL (equivalent to ~5 x 105 CFU) were then added to each well and the microplates were then incubated at 37 °C for 18 h. The MIC was defined as the lowest antimicrobial compound concentration that prevented growth of bacteria and was evaluated in three replicate assays.

To determine whether the observed inhibition was bacteriostatic or bactericidal, ten µL from wells with full inhibition (4 mg/mL IBP over 18 h) were plated on Blood agar or Luria Bertani agar and incubated for a further 18 h at 37 °C. Growth on the agars indicated the IBPs were bacteriostatic while the absence of any growth demonstrated they were bactericidal.

Ln 117. Results: The antibacterial capacity of the I. badionotus preparation (1-3 kDa BW) was tested against 5 x 105 CFU of Escherichia coli ATCC 25922, Pseudomonas aeruginosa ATCC 13637, Pseudomonas aeruginosa 64D (MDR), Staphylococcus aureus ATCC 25725, Staphylococcus aureus MR (Methicillin resistant), Enterococcus faecalis ATCC 51299 (Vancomycin resistant) and Enterococcus faecium 683D (vancomycin resistant). Following incubation for 18 h at 37 °C, no bacterial growth was observed with any of the tested bacterial species cultured in 4 mg/mL 1-3 kDa BW, variable growth was evident with 2 mg/mL 1-3 kDa BW, while expected growth occurred at 1-3 kDa BW concentrations of ≤ 1 mg/mL.

Ten µL of the bacterial suspensions that had been incubated for 18 h in 4 mg/mL 1-3kDa BW were then plated out on Blood agar or Luria Bertani agar and incubated at 37 °C for 18 h to evaluate viability. Significant growth was evident with Staphylococcus aureus ATCC 25725, Staphylococcus aureus MR (Methicillin-resistant), Enterococcus faecalis ATCC 51299 (Vancomycin-resistant) and Enterococcus faecium 683D (Vancomycin-resistant). In contrast, no growth was evident with Escherichia coli ATCC 25922, Pseudomonas aeruginosa ATCC 13637, or Pseudomonas aeruginosa 64D (MDR). These results indicate that, at a concentration of 4 mg/mL I. badionotus 1-3 kDa BW peptides were bacteriostatic against Gram-positive bacteria, but bactericidal to the Gram-negative bacteria tested.

Ln 316 Discussion: The antibacterial properties for I. badionotus 1-3 kDa BW digests were like those reported for Holothuria leucospilota, and other species (Ghanbari et al, 2012; Adibpour et al, 2014; Darya et al., 2020; Hossain et al, 2020).

  1. Lines 636–638; 667–669; 259–262: Methods mention “multiple comparisons and post-hoc tests” but specific tests, assumptions, and exact n for each dataset are not stated;

Figure 7 caption has an evident p-value asterisk error. Please specify tests (e.g., one-way ANOVA + Tukey), normality checks, n (biological vs technical), and correct captions. The error on this Figure has been corrected.

  1. Statistical approaches are briefly mentioned (ANOVA/post-hoc), but exact tests, number of replicates, and assumptions (normality, variance homogeneity) are not explained. Please clarify.

Ln 668. GraphPad Prism (https://www.graphpad.com/) was used to conduct the statistical analysis of data from each of three independent experiments. All data was expressed as mean ± SD and analysed by t-test, or by one-way analysis of variance (ANOVA) and Tukey’s post hoc test or the Bonferroni post hoc test for multiple comparisons if one-way ANOVA tests indicated statistical significance, which was set at ≤ 0.05.

  1. The wound-healing assay appears to be conducted with limited replicates. While six wells are mentioned, statistical robustness and independent repeats are unclear. Were assays repeated in different batches?

The wound healing assays were conducted three times, on each occasion with different batches of the 1-3 kDa BW, collagen 1-3 kDa, collagen 1-3 kDa Peak 2 preparations.

  1. The manuscript claims this is the “first” demonstration of collagen-derived peptides from I. badionotus in wound healing. However, related studies on sea cucumber species exist. The novelty should be more carefully contextualized against prior work.

Ln 293-298. Although this property has previously been reported for collagen peptides from several sea cucumber species [7], the present study is, as far as we can ascertain, the first to demonstrate that low-molecular-weight peptides of collagen from skin-free I. badionotus body wall promote wound healing and closure in vitro. The study also revealed that some non-collagen peptides from digests of I. badionotus body wall may have similar abilities to promote wound healing in vitro.

  1. The interpretation of gene expression is somewhat speculative. For example, upregulation of IL-6 and IL-1α is linked to positive healing, yet these are pro-inflammatory cytokines. Please provide stronger justification or reference support.

Ln 339-436. This issue is more clearly dealt within the modified Discussion

  1. Lines 386-400. The discussion suggests peptide-receptor interactions but presents no experimental evidence (e.g., receptor binding assays).

Ln 339-436. This aspect is based on reports in the literature on the potential modes of action of collagen peptides and indeed other peptides and was included as a possible explanation as to the significant differences in the actions of the three sea cucumber preparations evaluated in this study. A detailed examination of the mechanisms forms part of our ongoing research.

  1. Some steps in sample handling and collagen extraction are insufficiently detailed. For example, what was the yield of collagen? How reproducible were extraction batches? 

The animals used in the present study were captured from off the coast of Yucatan during the same seasons and had similar contents of body protein and collagen. Overall, the average yield of collagen from multiple batch preparations was 50 ± 10%

  1. Minor issues
  2. Line 2–4: Title is long; consider shortening for readability while keeping the key terms “sea cucumber,” “collagen peptides,” and “wound healing.”

We prefer to leave the title in its present form to emphasise to the reader that the findings within are not exclusive to collagen

  1. Line 39: Duplicate “Correspondence” word should be corrected.

This has been corrected.

  1. Line 46–47: “like, but not identical to” is colloquial; replace with “similar, though distinct from.”

Corrected.

  1. Line 62–64: “remarkable capacity for regeneration” sounds non-scientific; suggest a more formal phrasing.

Text substituted with ‘can fully regenerate’.

  1. Line 91–92: Collagen proportion (50%) should cite a specific reference or provide standard deviation/error.

Collagen proportion now given as 50 ± 10%

  1. Line 103–105: The phrase “significant wounds remained” is unclear; suggest “wound closure was incomplete.”

Corrected.

  1. Line 125–126: Clarify whether “10 µL” plating was in duplicates or triplicates.

Plating was done in triplicate.

  1. Line 138–139: Methods list is long and redundant. Consider condensing or grouping by type.

Sentence removed.

  1. Line 171–174: Please clarify whether the absorption band at 258 nm is consistently reproducible.

This band was consistent batch on batch.

  1. Line 199–202: The denaturation temperature (Td) is given without error margins. Please report mean ± SD.

Ln 201. Td =32.5 ± 0.2 °C

  1. Line 245: Cytotoxicity is described as absent, but higher doses were not reported. Please specify limits.

Ln 237. Before use in the scratch wound assay, the Collagen 1-3 kDa and Peak 2 fractions were tested for cytotoxicity by the standard MTT method. No toxicity was evident when HaCat cells were mixed with fractions at a concentration of 0.25, 0.125, 0.0625 and 0.0031 mg/mL. The concentration of 0.2 mg/mL was therefore used in the subsequent wound-healing assay.

  1. Line 300–304: Comparisons to other studies could benefit from a concise table summarizing species, composition, and Td values.

A table has been prepared and included as Supplemental data. Supplemental Table 1.

  1. Line 344–345: “Almost as good as” sounds informal; replace with “comparable to.”

Corrected

  1. Line 447–454: The animal collection process includes unnecessary detail (“plastic bags at depth”). This could be shortened.

Sea cucumbers can undergo extensive if mishandled or mismanaged during or after capture and transport to the research facility. For this reason, a detailed protocol for this phase is needed. However, this was described in detail in earlier papers by the group and these papers are now cited instead of the extended text.

  1. Several references (e.g., [16], [52]) are cited multiple times without clarity. Ensure consistency and update with most recent literature.

Several references, including these, deal with wound healing in vitro and gene expression promoted by external factors, such as light, and are included to provide comparisons with the changes observed during wound healing promoted by peptides from I. badionotus.

Round 2

Reviewer 1 Report

Comments and Suggestions for Authors

Dear authors and editors:

I thank the authors for their detailed responses and revisions. The effort to clarify study design and expand the discussion is appreciated. However, Percent match (27%) from iThenticate report might be relatively high. Please reduce it to around 20% to increase the scientific merits. While the authors have made progress addressing prior comments, I endorse the publication after minor revision and proofreading.

Sincerely,

Reviewer 2 Report

Comments and Suggestions for Authors

Although the reviewer’s report  identified certain limitations in the study, the revised version shows significant improvement and offers a contribution to the literature. Overall, the manuscript presents interesting findings. While these results could be strengthened by a more robust experimental design, the current version draws relevant conclusions based on the available dataset.